# Methodological Designs Applied in the Development of Computer-Based Training Programs for the Cognitive Rehabilitation in People with Mild Cognitive Impairment (MCI) and Mild Dementia. Systematic Review

**DOI:** 10.3390/jcm10061222

**Published:** 2021-03-16

**Authors:** Angie A. Diaz Baquero, Rose-Marie Dröes, María V. Perea Bartolomé, Eider Irazoki, José Miguel Toribio-Guzmán, Manuel A. Franco-Martín, Henriëtte van der Roest

**Affiliations:** 1Institute of Biomedical Research of Salamanca, University of Salamanca, 37001 Salamanca, Spain; mfm@intras.es; 2Department of Research and Development, INTRAS Foundation, 49001 Zamora, Spain; eider.irazoki@usal.es (E.I.); jmtg@intras.es (J.M.T.-G.); 3Department of Psychiatry, VU University Medical Center, Amsterdam University Medical Centers/Amsterdam Public Health Research Institute, Oldenaller 1, 1081HJ Amsterdam, The Netherlands; rm.droes@amsterdamumc.nl; 4Basic Psychology, Psychobiology and Methodology Department, Salamanca University, 37001 Salamanca, Spain; vperea@usal.es; 5Faculty of Psychology, University of Salamanca, 37001 Salamanca, Spain; 6Psychiatric Department, Rio Hortega University Hospital, 47012 Valladolid, Spain; 7Psychiatric Department, Zamora Healthcare Complex, 49071 Zamora, Spain; 8Department on Aging, Netherlands Institute of Mental Health and Addiction (TrimbosI Institute), 1013 GM Utrecht, The Netherlands; HRoest@trimbos.nl

**Keywords:** dementia, computer-based program, development design, cognitive training

## Abstract

In recent years, different computer-based cognitive training (CT) programs for people with dementia (PwD) have been developed following a psychosocial approach. Aim: This systematic review aims to identify the methodological designs applied in the development of computer-based training (CCT) programs for the rehabilitation of cognitive functioning in people with mild cognitive impairment (MCI) or mild dementia. Methods: A systematic review was conducted using the databases PubMed and PsycINFO. The search period was between 2000–2019. The study selection and data extraction processes were carried out by two independent reviewers. The protocol was registered in International Prospective Register of Systematic Reviews (PROSPERO) under registration number CRD42020159027. Results: Thirteen studies met the inclusion criteria. The most frequently used methodological design in the development of CCT programs for people with MCI or mild dementia was the user-centered design (UCD). This design involves an interactive system characterized by the inclusion of end users from the initial stages of its development, throughout the establishment of functional requirements, and in the evaluation of the program’s usability and user-experience (UX). Conclusion: UCD was the most used methodological design for the development of CCT programs although there was quite some variation in how this design was applied. Recommendations for future studies about the development of CCT programs for people with MCI and mild dementia are given. Central focus should be the inclusion and active participation of end users from the initial stages of development.

## 1. Introduction

Dementia is a neurodegenerative disease characterized by cognitive, emotional, and social deficits. Cognitive functional impairments generally involve problems with orientation, memory, attention, language, reasoning, calculation, and executive functioning. Emotional alterations, for example, are associated with symptoms like anxiety, depression and apathy. As a result of these cognitive and emotional alterations, people suffering from dementia may have social problems, such as difficulties participating in their social network and, consequently, loss of social contacts. Currently, it is estimated that there are more than 50 million people with dementia (PwD) in the world, and, by 2050, this number is expected to have increased to 152 million PwD [1].

Pharmacological therapy is the most common treatment for this disease. A large number of clinical trials are being carried out continuously to test the effectiveness of different drugs that help counteract some of the symptoms of the disease in the short term [2,3,4]. However, as long as there is no cure for dementia, pharmacological therapy will only be part of the care and treatment for PwD.

In recent years, from a psychosocial–technological approach, different cognitive rehabilitation programs have been used with PwD as a complement to medication. Some existing computer-based cognitive training (CCT) programs for people with mild cognitive impairment (MCI) and mild dementia have been flagged by systematic reviews [5,6]. However, when reviewing the studies focusing on them, we found that they failed to provide a detailed description of the methodological design used in the development of these CCT programs for people with MCI and mild dementia. Some of the studies [7,8,9], including those associated with the BRAINER; CogMed; SOCIABLE; Kitchen and Cooking programs; and the websites of the CogMed, CogniPlus, FesKits programs, and SOCIABLE, only reported the participation of an interdisciplinary group (neuropsychologists and game designers) in the development of the programs, providing no further details. Some of them, such as CogMed, had even been originally developed with people affected by other disorders [10].

Consequently, studies have been conducted to investigate the usability [11,12,13] and effectiveness of such programs in terms of maintaining cognitive performance and delaying cognitive decline in PwD [14,15,16,17]. Nevertheless, many of these programs have never been implemented in clinical practice.

Because of its impact on use, design plays an important role in the development of any computer program and should therefore consider parameters like user-friendliness, being simple, clear, and easy to use [18]. Usability is one of the main criteria that must be fulfilled by a program [19], which should be challenging and useful to motivate the user, poor usability being a demotivating factor [20].

ISO9241–210 [21] proposes a series of standards that should be taken into consideration when developing a human-centered design (HCD) program. These standards are as follows: (1) defining the context (CTX), (2) specifying user requirements, (3) designing, and (4) evaluating the design. The design should start from and respond to users’ current needs, involving them in its methodology by means of, among other things, observation (OB), interviews/questionnaires (Q), and field tests [22].

The design of any computer-based program should take into account the characteristics and needs of the target population [23,24]. Older adults may present sensory disturbances associated with hearing and visual impairments, which make it difficult to carry out certain perceptual activities. They may also have physical difficulties affecting their mobility and, more specifically, their fine motor skills. The type of deterioration and its severity vary from one person to another, which also applies to people diagnosed with dementia [25].

In response to these needs, the design of cognitive rehabilitation programs should include the use of appropriate colors, size and text font, background style and sounds [18], touch screens [26,27], and different exercises and cognitive levels [28] to enhance usability and so that the program may be accessible for older adults. Moreover, it is necessary to involve PwD in the design process [29] to know how users experience the computer-based program [30] since a positive or negative impact could determine whether the technology is adopted or not.

In line with the mentioned principles, this systematic review aims to identify the methodological designs used in the development of CCT programs for the rehabilitation of cognitive functioning in people with MCI and people with mild dementia and to describe the ISO9241-210 [21] standards that were followed in the different studies for the development of the computer-based programs.

## 2. Materials and Methods

### 2.1. Systematic Literature Review

A systematic review was conducted using the PubMed and PsycINFO databases. The protocol was based on the Preferred Reporting Items for Systematic Review and Meta-Analysis-PRISMA statements [31] and registered in International Prospective Register of Systematic Reviews (PROSPERO) under registration number CRD42020159027 or EC626091945 in April 2020 and updated in October 2020 [32].

### 2.2. Search Strategies

The databases were searched from inception onwards by the first author under the supervision of a medical information specialist of the VU University Medical Center, on 5 November, 2019. The search terms used were: dementia-MCI AND computer-based program AND development or design AND cognitive. The search period ranged from January, 2000 to October, 2019.

### 2.3. Eligibility Criteria

Papers targeting people with MCI or mild dementia (Alzheimer’s, vascular, frontotemporal dementia) and describing the development process, design, use and/or results of a pilot study, or effect of a study associated with CCT programs were selected, and only papers published in peer-reviewed journals between January, 2000–October, 2019 and written in English or Spanish were accepted.

The review focused on the most prevalent types of dementia (Alzheimer’s disease (AD), vascular dementia (VD), mixed AD-VD), excluding those that are less common, such as Lewy bodies, Pick’s disease, Huntington, or Parkinson’s. Studies that did not include the development of CCT programs and whose main and only topic was the effectiveness or usability of a CCT program were also excluded.

Systematic reviews and meta-analyses were considered for review with the purpose of finding studies that met the inclusion criteria and had not been identified by the initial search strategy.

### 2.4. Selection Process

The search results from the two databases were uploaded into EndNote X7. Duplicate studies were identified and removed according to the digital guidelines of the VU University Library Amsterdam, the Netherlands [33]

After the process of removing duplicates, two independent reviewers (AADB, EI, MAFM, and HVDR) assessed the titles and abstracts of every study to identify potential studies that met the inclusion criteria. The reviewers considered the additional information, consulting the studies online. Any disagreements between the reviewers regarding the included and excluded studies were discussed until consensus was reached.

The full texts of the included studies were obtained in order to perform a critical reading and extract the relevant information. This process was carried out by AADB. Some systematic reviews were selected for review based on the references included, leading to the addition of new studies that had not been tracked by the initial search strategy. The list of references of these systematic reviews or meta-analyses was examined by one independent reviewer (AADB).

### 2.5. Data Synthesis

A critical analysis was conducted to identify the methodological designs applied in the development of CCT programs for cognitive function rehabilitation in people with MCI and mild dementia.

First, we extracted information associated with the characteristics of the rehabilitation programs (name of the program), characterization of the sample (sample size, sample distribution by sex, study groups, and diagnosis), country where the data collection was carried out, and dropouts (number and main reasons).

Subsequently, the following information was extracted: (a) type of methodological design applied by each study for the development of the CCT programs in people with MCI and mild dementia and (b) how the ISO9241-210 (21) standards were followed throughout the development of the programs, i.e.,: (1) understanding and specifying the CTX of use, (2) specifying user requirements, (3) producing design solutions, and (4) evaluating the design. The baseline study requirements, measurement instruments, and main results were also extracted.

### 2.6. Study Selection

The study selection process is presented according to the PRISMA Flow Chart [31] (Figure 1). The search strategy yielded 190 studies (63 studies in PubMed and 127 studies in PsycINFO). Four papers tracked from two publications were added. After removing duplicates, 182 studies were reviewed for their title, abstract, and additional information obtained online.

One hundred and fifty-two studies were excluded due to not meeting the inclusion criteria (the reasons for exclusion are detailed in the PRISMA Flow chart, Figure 1), leaving 30 eligible studies. After reviewing the full text of these 30 studies, 17 were excluded on the following grounds: 10 did not fulfill the inclusion criteria, the full text of six was not available, even though we contacted the main author (who did not respond to our request), and one study was a systematic review. This systematic review was checked against its reference list to find potential papers, and six possible studies were identified and discarded after reviewing their abstracts.

Finally, only 13 studies met the inclusion criteria and were included in the systematic review. Of these 13 studies, the paper by Haesner, Steinert [34] appeared to be a continuation of a study that was reported in another selected paper by Haesner, O’Sullivan [35]. Three studies, Ben-Sadoun, Sacco [36], Benveniste, Jouvelot [37] and Boulay, Benveniste [38], were identified from the study of Ben-Sadoun, Manera [39]. Particularly, the study by Boulay, Benveniste [38] was the continuation of the study by Benveniste, Jouvelot [37]; this latter talks about the user requirements and initial development phases while the first study talks about the evaluation of the final prototype of MinWii, respectively.

**Figure 1 jcm-10-01222-f001:**
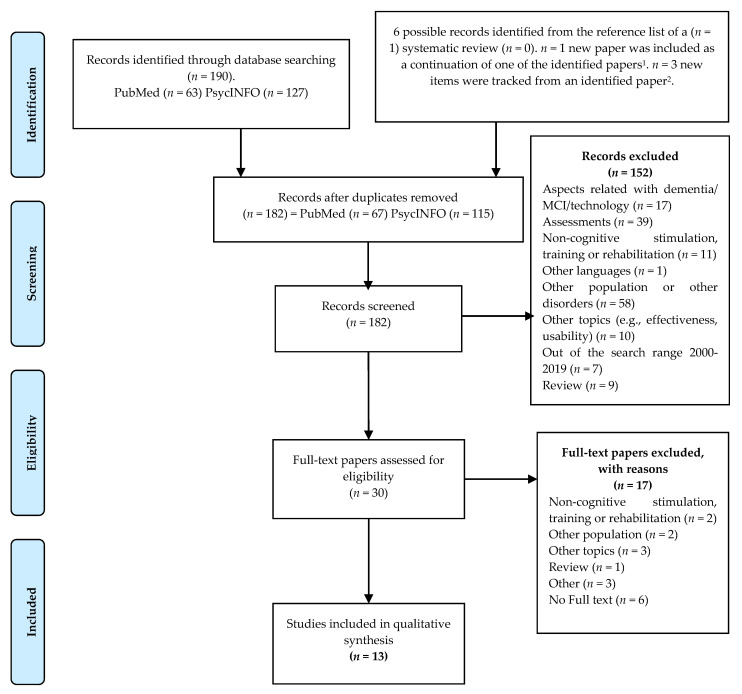
PRISMA Flow Chart. Selection process of the systematic review (From: Moher, Liberati [31]). ^1^ Haesner, O’Sullivan [35]. ^2^ Ben-Sadoun, Manera [39].

## 3. Results.

### 3.1. Study Aims

A total of 11 CCT programs described in 13 different studies were identified (Table 1). The aim of the studies identified in this review was to develop CCT programs or games for people with MCI or mild dementia, including the user from the initial stages of their development, design, and prototype assessment, and conducting usability and user-experience (UX) assessments throughout the development of each of the prototypes.

### 3.2. Participants, Settings, and End Users

Participant characteristics and settings are presented in Table 1. Two (15.4%) studies included two samples with different purposes: Scase, Kreiner [46] included a sample of 18 people with the objective of evaluating the UX of a pre-prototype of the games and a second sample of 25 people to evaluate the UX of the final prototype of the games. The use of two samples had similar purposes in the study carried out by Scase, Marandure [45]; although, this study included a sample of 18 people to evaluate the pre-prototype and 24 people to evaluate the UX of the final prototype and adherence to the games. On the other hand, four (30.8%) studies associated with two programs (LeVer learning platform and MinWii) also used different samples. Specifically, Haesner, O’Sullivan [35] recruited 12 people to investigate the requirements that should be considered in the development of an online platform for cognitive training (CT). These requirements were taken into consideration in a second study carried out by Haesner, Steinert [34], which included 80 people to assess the usability and UX of the final LeVer learning platform. Equally, the Benveniste, Jouvelot [37] study recruited nine people to indicate user requirements, and Boulay, Benveniste [38] recruited seven people for the evaluation of the final MinWii prototype. Sample sizes varied from study to study. In general, the studies of Cruz; Pais J. [40]; Gao [22]; Haesner, Steinert [34]; Scase, Kreiner [46]; and Scase, Marandure [45] used a large sample size (≥30) while the others used a small sample size (≤30).

Regarding the diagnosis of the sample population of the studies included in this review, seven (53.9%) studies recruited people diagnosed with MCI [34,35,41,42,43,45,46], four (30.8%) studies worked with people diagnosed with mild dementia [22,37,38,44], and two (15.4%) studies included a pooled sample of people diagnosed with MCI and mild dementia [36,40]. Four (30.8%) studies included healthy comparison subjects [34,35,36,44], two (15.4%) studies included people with other types of diagnoses in their sample (e.g., traumatic brain injury (TBI) or Lewy body dementia (LBD)) [40,44], and one (7.7%) study worked with people with moderate and severe stage dementia [43] (Table 1).

The 13 studies selected for this systematic review included a total of 400 people, of whom 45.8% (*n* = 183) were people with MCI, 24.3% (*n* = 97) were people with mild dementia, 16.5% (*n* = 66) were healthy people, 2.5% (*n* = 10) were people with MCI or mild dementia, 0.8% (*n* = 3) were people diagnosed with moderate dementia, and 10.3% (*n* = 41) were people with other disorders (e.g., TBI). Of the 13 included studies, 30.8% (*n* = 4) reported dropouts in their samples. The study by Cruz, Pais J. [40] had a dropout rate of 40%; Han, Oh [41] reported a 30% dropout rate; Otsuka, Tanemura [42] reported 26.3% dropout; and Boulay, Benveniste [38] had a 28.6% dropout rate. The main reasons for dropout were failure to regularly attend the training sessions, difficulty in using the device, development of psychiatric disorders, visual impairment, and hospitalization (Table 1).

The people diagnosed with MCI or mild dementia included in the studies were between 50–88 years old. The healthy people were between 60–85 years old. However, some of the studies failed to mention the mean age and/or standard deviation of their sample [22,34,35,37,38,41,43] (Table 1).

Of the 13 studies included in this review, four (30.8%) recruited their sample from hospitals [35,37,38,41], two (15.4%) from memory clinics [36,40], two (15.4%) from day centers or care centers [42,43], one (7.7%) from nursing homes [22], two (15.4%) from medical centers or institutions associated with universities [34,44]*,* and two (15.4%) from other centers [45,46] (Table 1).

### 3.3. Methodological Design Used for the Development

Table 2 describes the specification of the methodological design used by each of the studies in accordance with the international standards proposed by ISO9241-210 [21] for the development of programs: (1) understanding and specifying the CTX of use (type, characteristics and tasks of users, and physical or social environment), (2) specifying the user requirements, (3) producing design solutions, and (4) evaluating the design.

Of the 13 studies included in this systematic review, 11 (84.6%) used a user-centered design (UCD) [34,35,36,37,38,41,42,44,45,46] or HCD [43] for the development of the cognitive rehabilitation programs or games. According to ISO9241-210 [21], UCD and HCD are equivalent terms. On the other hand, one study (7.7%) used a human–computer interaction design (HCI) [22], and one study used an end-user interaction design [40] (Table 3).

#### 3.3.1. Understanding and Specifying the CTX of Use

Four (30.8%) studies were based on observation of their users [36,37,38,42] and all the studies (*N* = 13) reviewed the literature as a means to understand the CTX of use (Table 2) in terms of type, user characteristics (or needs and deficits) and tasks, and physical or social environment of people with MCI or mild dementia. Accordingly, all studies specified the characteristics of the users or target population and the CTX of use of the program or game to be developed, i.e., the application environment and tasks (Table 1).

#### 3.3.2. Specification of User Requirements

Six (54.5%) studies designed or used a pre-prototype of the program in order to specify target population requirements [35,37,41,43,45,46], in contrast with five (45.5%) studies that did not specify them and did not use a pre-prototype of the program [22,36,40,42,44] (Table 2).

Pre-prototypes were evaluated through interviews in two (33.3%) studies [45,46], based on questionnaires in one study (16.6%) [43], using a combination of interviews and questionnaires in one study (16.6%) [35], and through OB in one study (16.6%) [37], one study (16.6%) failing to specify the method used [41] (Table 2).

Scase, Marandure [45] and Scase, Kreiner [46] conducted focus group (FG) interviews with people with MCI using a pre-prototype of their games and gamified environment in the Decrease of cOgnitive decline, malnutRition, and sedEntariness by elderly empowerment in lifestyle Management and social Inclusion (DOREMI) Project. This study identified the following baseline requirements: prior knowledge of computer use; preferences for games such as puzzles, cards, and quizzes; the importance of adapting the interface design according to physical problems; the use of clear instructions; and the inclusion of challenging games (Table 3).

Haesner, O’Sullivan [35] used a semi-structured interview and a questionnaire, alongside existing programs on the web to identify the baseline requirements. In other words, they did not build a pre-prototype but built their own prototype from existing prototypes. Thus, they reported the following baseline requirements: (1) attractive looking design, (2) face-to-face training course, (3) pretest followed by progress assessments, (4) exercises should include tasks and challenges associated with the real life of the people and be of interest to them, (5) inclusion of personal comments/feedback in real time, (6) the training situation should help improve the person’s performance, (7) possibility of communication through audio-video between people, and (8) additional information for self-training (Table 3).

The starting point of Han, Oh [41] was the limitations found in an existing program: Spaced Retrieval-based Memory Advancement and Rehabilitation Training (SMART). These limitations, which included the need for the constant presence of a professional during the sessions, entailing high costs and, therefore, the impossibility of increasing treatment intensity, became the main focus for the design and development of a new program: The Ubiquitous Spaced Retrieval-based Memory Advancement and Rehabilitation Training (USMART) (Table 3).

The purpose of Peeters, Harbers [43] was to develop a Music ePartner for the rehabilitation of episodic memory in people with MCI. Hence, a questionnaire was used to evaluate people’s musical preferences so that the authors could customize the design to users’ preferences and needs (Table 3).

Benveniste, Jouvelot [37] found different baseline requirements for the development of the MinWii: (a) the use of a Wiimote system to allow greater accessibility for people with motor problems; (b) the use of a graphical interface to counteract visual problems; (c) the use of simple games to reduce feelings of frustration; (d) simplicity of design and the use of familiar devices such as a TV or computer (Table 3).

#### 3.3.3. Development of Final Prototype

Eleven (84.6%) studies developed a final prototype of the program [22,34,36,38,40,41,42,43,44,45,46]. Of these, 10 (90.9%) studies used qualitative methods aimed at evaluating the final prototype of the program, and one study (9.1%) did not report an evaluation of the usability or UX aspects of the final prototype [22] (Table 2).

#### 3.3.4. Evaluation Methods

From the studies (*n* = 10) that evaluated the final prototype of the program, three studies evaluated UX [42,45,46] and seven evaluated usability and UX [34,36,38,40,41,43,44]. Three studies evaluated the intervention through interviews [43,45,46], one used a combination of OB and interviews [44], one used a combination of OB and a questionnaire [38], and five used questionnaires [34,36,40,41,42] (Table 2 and Table 3).

#### 3.3.5. Usability

Music ePartner’s intuitive interface used touch screens that made it easy to use, and the inclusion of personalized settings in the interface [43] expresses the preferences of the end user [36].

Particular aspects of the interface of each program that were highlighted by users were, for example, the clarity of the instructions in COGWEB [40] and the use of plain and straightforward language in the games Wizard-of-Oz (WoZ) interface, which facilitated better understanding [44].

Haesner, Steinert [34] also took different aspects of usability into consideration, administering a questionnaire to end users, who had the opportunity to interact with the LeVer learning Platform. The exercises achieved high acceptance rates (97.3%): 90% of the people rated the graphic design and content positively, 97.4% rated usability positively, and 36.9% considered the platform very useful. The content (12.7 ± 3.1 out of 20 points), accuracy (7.4 ± 1.5 out of 10 points), layout (7.3 ± 1.5 out of 10 points), ease of use (7.1 ± 1.7 out of 10 points), timeliness (6.6 ± 2.0 out of 10 points), and program speed (9.4 ± 2.9 out of 15 points) on USMART also received positive ratings [41].

#### 3.3.6. User-Experience

People with MCI [43,45,46] and PwD [36,38,44] rated their experience positively regarding the use of the final program. Users found the exercises included in the program COGWEB [40], DOREMI project [45], and Music ePartner [43] interesting. Eleven out of thirteen healthy old people (HOP) and 8/10 PwD enjoyed the activities in the games of WoZ interface [44]. Likewise, people with MCI who used Atama-no-dojo [42] and Music ePartner [43] enjoyed the exercises of each program, respectively. Additionally, the USMART program received a satisfaction score of 8/10 from people with MCI [41].

The “Find it”, “Match it”, and “Solve it” games in the DOREMI Project [46], Quiz and Prover tasks in the WoZ interface [44] and the X-Top games [36] were all highly appreciated by users, who regarded them as challenging and interesting. However, the “Complete it” game in the DOREMI Project [46] and the sorting activity in the WoZ interface [44] received negative feedback from users, who reported that their low difficulty levels generated feelings of boredom.

In addition, the people with MCI participating in the studies selected for this review pointed out that using these programs had benefited their cognitive performance [34,42,45,46], specifically their memory (71.4%) [41]. They also indicated the physical [45], emotional, and social benefits obtained, among which were that they contributed to promote interaction with their caregivers [43] and other people [38,45].

Peeters, Harbers [43] differentiated between people with different levels of cognitive impairment, noting that people in the early stage of dementia often gained more benefits from using CT programs than people in advanced stages of the disease. Boulay, Benveniste [38] also mentioned that the level of cognitive impairment influenced the use of the Wiimote, even though users were able to learn how to use it.

People with MCI expressed interest in purchasing and continuing to use the USMART program autonomously at home [41]. In the Haesner, Steinert [34] study, 89.2% of the people mentioned that they would continue using the LeVer learning platform.

### 3.4. Risk of Bias in Outcome Assessment

We consider that the PRISMA “risk of bias outcome assessment” item is not applicable to this review, because our objective was to identify the methodological designs and methods applied in the development of CCT programs for the rehabilitation of cognitive functioning in people with MCI and mild dementia, and not to evaluate the effectiveness of these methodological designs or the developed products.

## 4. Discussion

In the past years, different CCT programs have been developed [47,48], targeting people with different pathologies, including MCI and mild dementia [49]. These programs aim to preserve or improve cognitive abilities [50,51,52]. However, to promote their implementation, dissemination, and successful adoption by their target users, it is important that such users be involved throughout the development process. To gain insight into the extent to which this has been so in the development of CCT programs for people with MCI and mild dementia, we conducted this systematic review aimed at identifying the designs and methods applied in the development of these CCT programs.

We identified 13 studies on the development of CCT programs, which described 11 different CCT programs. All the studies included for this systematic review used some variant of end-user centered methodological design. Likewise, most studies fulfilled most or all of the criteria of the International Standards for HCD for Interactive Systems [21]. These standards provide an approach to the development of interactive systems that are usable, efficient, ergonomic, accessible, sustainable, and safe for the end-user.

*Understanding and specifying the CTX* of use is one of the main standards proposed by the ISO9241-210 [21]. The studies included in this systematic review used OB and literature review as a source to understand the CTX of use. All of them specified the CTX by reporting the characteristics and needs of the program’s target population, as well as the environment in which it would be applied and the tasks the users were to perform.

*Specification of user requirements*. Only half of the studies took this standard into account. The use of a qualitative methodology was predominant, as was the development or use of pre-prototypes to define user requirements. Previous knowledge or use of the computer in the elderly was a factor that was taken into account in the development of CCT programs [45,46]. This requirement stems from the little familiarity that older people have with technology, as indicated in previous studies [53]. Therefore, interfaces need to fulfill at least some minimum requirements, such as an intuitive, graphical, simple, familiar [37], and attractive interface [35]. Nevertheless, despite these findings, Góngora Alonso, Toribio Guzmán [13] pointed out that lack of experience in the use of technology does not necessarily have to influence its use.

Our study also demonstrates the need to adapt and develop a CCT program involving end users from the initial stages, taking into consideration their physical or motor difficulties [45,46], which could be one of the main reasons for a person to stop using the CCT program [41,42]. The importance of considering this need in the development of CCT programs was also pointed out in and consistent with other studies, such as the study of Góngora Alonso, Toribio Guzmán [13]. In addition, our review also demonstrated the importance of taking into account cognitive problems, including those related to language [54], for example, by using simple and clear instructions, as noted by Scase, Marandure [45]; and Scase, Kreiner [46].

Furthermore, our findings pointed out some of the specific requirements that CCT program exercises should have. Although some approaches seem contradictory, this does not mean that they are not valid and necessary, an example being how Benveniste, Jouvelot [37] states that the exercises should be easy to avoid frustration, whereas Scase, Marandure [45] posed challenging exercises to maintain motivation. The right answer probably lies somewhere in between: the programs should include conservative approaches associated with exercises that can be adapted to users’ cognitive level without them finding them too easy or too difficult. Another important requirement, brought forward by Han, Oh [41], is the capacity of modifying treatment intensity according to the person’s performance. This is an important finding that has also been pointed out in other studies, such as that by Franco-Martin, Diaz-Baquero [55].

Other studies extensively discussed the ecological validity of the exercises. Djabelkhir, Wu [56] suggested that people with MCI integrate a technological device (Tablet–PC) into their daily lives. Although the concept of ecological validity is not very recent, it is still little applied in CCT programs. Nevertheless, the study by Haesner, O’Sullivan [35] does mention its relevance. Ecological validity encourages thought and reflection throughout the early stages of development on how the exercises included in a CCT program could contribute or transfer to users’ real life, i.e., that exercises have a practical application for people.

Another requirement was the importance of including real-time feedback [35]. Feedback has also been described by other studies as a basic training principle that helps maximize the benefit of a CCT program [57].

In general, the specification of user requirements mentioned by the studies included in this systematic review took into account the needs, experience, and knowledge of the end users [58], which positively contributed to the design of the final programs. In particular, these requirements were associated with the changes and characteristics (physiological, neuropsychological, social, and physical changes) of older PwD mentioned by Guisado-Fernández, Giunti [18] in his systematic review. However, our review also shows that 45.5% of the studies did not report such baseline requirements. We consider that the establishment of user requirements based on end-user participation from the early stages of the development is of great importance in the development of any CCT program because this will allow the creation and development of more usable and personalized CCT programs [29]. Ideally, the interface should be adjustable to the specific functional and cognitive impairments of the older adults with MCI and mild dementia who are to use it [28,59].

*Design evaluation*. Overall, most of the identified studies took usability and UX evaluations into account from the use of the same sample. However, the Scase, Marandure [45] and Scase, Kreiner [46] studies considered the use of a parallel or second sample for this phase. Undoubtedly, the above represents a methodological advantage because it provides greater objectivity and eliminates biases. In this order of ideas, usability mainly focused on the concepts of familiarity and ease of use. Lack of usability has been considered one of the most important demotivating factors in the use of CCT programs [20]. Usability is regarded as a fundamental and predisposing factor for the use and adoption of a CCT program, and most of the studies included in this systematic review used a touch screen navigation system, which turned out to be more intuitive and usable for people with MCI and mild dementia with few technological skills.

The COGWEB [40] and WoZ Interface [44] programs stood out for their use of clear instructions and simple and straightforward language. These findings are in line with the approach of Tziraki, Berenbaum [60], who point out the importance of using a clear semantic structure for instructions. In general, usability aspects should also encompass the interface’s graphic design, as well as the content and exercises [34]. Additional aspects are accuracy, ease of use, timeliness, and program speed [41].

Furthermore, 90% of the studies included UX evaluations of their final program. UX is a subjective aspect that influences the acceptance of a CCT program by its end users. In general, our review shows that people with MCI and mild dementia positively evaluated the exercises included in each program. However, levels of satisfaction and interest in the DOREMI project [45,46], WoZ Interface [44], and X-top [36] programs depended on the level of difficulty of the exercises. The inclusion of challenging activities has also been identified as one of the motivational factors for the use of CCT programs [20].

People with MCI and mild dementia mentioned that they benefited from the CCT programs at the cognitive, emotional, physical, and social levels [34,38,41,42,43,45,46]. Expectations regarding the use of a CCT program are often associated with the benefits and support they can potentially provide in relation to specific aspects of cognitive functioning [61]. The estimated potential benefit is a key aspect, since it influences the motivation to continue using the program [62]. However, sometimes these expectations are unrealistic and extreme; therefore, the person must be well informed about the scope and limitations of a CCT [18].

Boulay, Benveniste [38] and Peeters, Harbers [43] both pointed out that the level of deterioration suffered by the people influenced the benefits they perceived from the use of the CCT program. This was confirmed in other studies, which reported that PwD with higher levels of impairment could find difficulties in using technology and therefore required closer supervision and continuous repetition of instructions [62]. This may impact their self-efficacy [63,64] and cause lower motivation to continue using a certain program. On the other hand, Hofmann, Hock [65] showed that, after three weeks of training, people with AD became faster in executing the tasks and needed less and less help using the computer. This last finding was similar to that reported in the approach of Boulay, Benveniste [38].

### 4.1. Limitations

Some studies that fulfilled the inclusion criteria may not have been detected due to limitations in our search strategy or the small number of databases used for this review. However, taking into account our objective, we consider the databases used the most appropriate due to their clinical orientation. Therefore, we did not use additional specific technological databases as these are generally more oriented to the advancement of the technology from an engineering point of view and not so much from a clinical perspective. Nevertheless, Future studies are recommended to expand the search strategy by using additional databases. Another limitation of our study was that we were unable to retrieve the full text of six studies that met the criteria after reading the abstract, because the full text was not accessible from various databases or via the corresponding author. It is also possible that some recent publications have not (yet) been included in the selected databases.

Furthermore, gray literature was not included in the search because we considered the topic of methodological design in the development of the program to be complex and relatively new, especially when applied to people with MCI and mild dementia.

### 4.2. Recommendations

Whereas the definition of CTX of use is important as a first phase, it is clearly not enough to design a CCT program, which requires the creation and assessment of a pre-prototype that involves end users from the initial stages of development to find out their requirements. Identifying these requirements does not only tackle most of the methodological problems, but also those associated with the quality and use of the final program, while also avoiding future additional costs resulting from subsequent necessary adaptations. It is therefore strongly recommended that future studies use an interactive and participatory design, including end users from the beginning of the pre-prototype development, carrying out evaluations in order to identify user requirements and, in turn, including them in the final development of the prototype.

Regarding the methodological phase of the evaluation of the final prototype, usability and UX tests generally added quality to the programs, described the user’s adaptation, and determined the impact, use, and significance of the CCT program. It is recommended that future studies include both types of evaluations (usability and UX), as both provide complementary and important data to consider in the development of a CCT program for people with MCI and mild dementia.

The wide heterogeneity in the application of the user- or human-centered methodological design for the development of CCT programs in people with MCI and mild dementia was evident. Therefore, our findings indicate the need to apply this methodology in a more standardized way in the design of CCT programs, based on a series of clear rules that this field of research can follow, such as ISO9241-210 [21].

The scientific value of this review is that it summarizes the evidence regarding the methodological designs used in the development of CCT programs in people with MCI and mild dementia and the ways in which different studies followed the ISO9241-210 (2019) standards [21] for the development of CCT programs. We propose that better descriptions of the used methodological design are included in future publications, as well as a critical reflection regarding the design and development process of CCT programs. Few studies have been carried out on this topic in this field of study, and most systematic reviews focus on the evaluation of the effectiveness of CCT programs.

This systematic review also has practical value for developers because it provides recommendations related to the different standards proposed by the ISO9241-210 [21] for the development of programs from a UCD. These recommendations and findings should be taken into consideration in future research or projects that seek to develop a CCT program for people with MCI and mild dementia, with the aim of minimizing errors in program development, thus avoiding higher long-term costs and generating maximum effectiveness of the intervention.

Finally, following these recommendations in the development of CCT programs will also provide value in the clinical field. That is, the clinical field of Neuropsychology will have more usable CCT programs and, in turn, more effective programs according to their therapeutic objective, for example, to maintain cognitive functioning and delay cognitive deterioration. Above all, this field will have CCT programs designed and developed from and for people with MCI and mild dementia. The target population should be the central and starting point of any CCT program.

## 5. Conclusions

Taking into consideration the objective of this systematic review, we conclude that UCD was the most used methodological design for the development of CCT programs. However, we found variations in its use and application across studies. Some of these variations suggested possible flaws in the design and might have led to problems associated with the usability and UX of these programs.

We propose the use of a user-centered methodology for the development of CCT programs for people with MCI or mild dementia. This methodology is to be regarded as an interactive process where the inclusion and active participation of the end-user from the initial stages of design is the central focus [66]. This will lead to higher usability and better UX, avoiding additional costs for future adaptations and increasing the likelihood of adaptation of the program and opportunities on the technology market. Any study that intends to develop a CCT program for people with MCI or mild dementia should take UCD standards into account.

Finally, we underline some key points and/or objectives that should be taken into account for the development of a CCT program for people with MCI and mild dementia for each of the criteria established by the ISO9241-210 [21]:(a)Understand and specify the CTX to use:Describe and characterize the target population (MCI and mild dementia): sociodemographic (age group, sex, educational level) and clinical (diagnosis, physical, psychological, cognitive symptoms) aspects.Define the CTX of use: day center, nursing home, clinic, home, etc.(b)Specify user requirements:Involve end users from this stage.Develop a pre-prototype CCT program.Use of qualitative methodology: OB, interviews, questionnaires, etc.Inquire about the physical and sensory (visual or auditory disturbances), social (social red), and cognitive (orientation, attention, memory, language, executive function, calculation alterations) needs of the user and their impact on the use of the CCT program.Inquire about previous knowledge about the use of technological devices and the preferences of the elderly.Define and propose the program requirements with respect to design (device type and navigation method) and content (type of exercises, difficulty levels, type of instructions and feedback) so that these adjust to those specific needs.(c)Program design and evaluation:Involve end users.Use of qualitative methodology.Evaluation of usability regarding the design (intuitive, graphical, simple, familiar) and content of the CCT program.Evaluation of the UX regarding the level of satisfaction, experience (positive or negative), and expectations of the people with MCI and mild dementia.

## Figures and Tables

**Table 1 jcm-10-01222-t001:** Sample characteristics for each study.

Program	Study	Participants/Diagnosis	Settings	Drop-Outs
**MinWii**	Benveniste, Jouvelot [37]	*N* = 9. Dx: AD (MMSE: 10–25).	Geriatrics Unit of Hospital Saint-Maurice (France).	None.
**MinWii**	Boulay, Benveniste [38]	*N*= 7. 3 males. Mean age: 88.5. Dx: AD (MMSE 12–30).	Long Term Care Unit at La Collégiale Hospital (Paris, France).	*n* = 2. 1 medical problems. 1 refused to continue.
**COGWEB**	Cruz, Pais J. [40]	*N* = 48/80. 21/48 females. Mean age: 60 ± 13.3. Mean education: 6 ± 4.3. Dx: 25%: subjective memory complaints. 25%: TBI. 25%: stroke and other static brain lesions. 25%: mild AD.	Outpatient memory clinic (Portugal).	*n* = 32 (not able to attend the assessment session at the hospital).
**USMART**	Han, Oh [41]	*N* = 10. 1 males/9 females. Dx: n= 4 aMCI single-domain type, *n*= 2 aMCI multiple-domain type, *n*= 1 naMCI single-domain type.	KLOSCAD. Dementia Clinic of the Seoul National University Bundang Hospital (South Korea).	*n* = 3: 1 difficulty learning how to use the iPad. 1 felt the inter-retrieval activities were childish. 1 developed an acute physical illness.
**LeVer learning platform**	Haesner, O’Sullivan [35]	HOP Group: *N*= 6. 3 males. Age: 60–70 years. Education: n = 1: < 9 years; *n* = 2: 10–13 years; *n* = 3: university degree. MCI Group: *N*= 6. 3 males. Age: over 80 years old. Education: *n* = 3: < 9 years; *n*= 2: 10–13 years; *n* = 1: university degree.	Geriatric hospital (Charité Berlin).	None.
**LeVer learning platform**	Haesner, Steinert [34]	*N* = 80. 44 females. Mean age: 70 years. Dx. *n* = 39 HOP. *n*= 41 MCI.	Senior University, Berlin (Germany).	None.
**Software “Atama-no-dojo”.**	Otsuka, Tanemura [42]	*N*= 19 females. Mean age: 82.2 ± 2.9. Mean education: 10.5 ± 1.7. Dx: MCI (MMSE ≥ 24).	A day-service center in Hyogo Prefecture (Japan).	*n* = 5: 1 vision problem due to macula degeneration, 1 hospitalized with lung cancer, 3 who attended less than 80% of our program sessions.
**X-Top**	Ben-Sadoun, Sacco [36]	MCI and AD Group: *N* = 10. 6 males. Mean age: 82.3 ± 6.4. HOP Group: *N*= 8. 3 males. Mean age: 71.4 ± 10.1.	Memory Center in Nice (France).	None.
**Music ePartner**	Peeters, Harbers [43]	*N* = 5: 1 male; Age: 60 years. Dx: MCI. 1 male; Age: 70 years; Dx: PwD Moderate–severe. 1 male; Age: 80 years; Dx: PwD Severe. 1 male; Age: 70 years; Dx: PwD Moderate. 1 male; Age: 50 years; Dx: MCI.	Care Organization Pieter van Foreest, Zuid-Holland, (the Netherlands).	None.
**Games. (WoZ) interface**	Dethlefs, Milders [44]	*N* = 23. HOP Group: *N* = 13. 9 males/4 females. Mean age: 84.33 years. PwD Group: *N*= 10. 8 males/2 females. Mean age: 78.20 years. Dx: *n* = 7 AD, *n* = 2 early onset dementia, *n* = 1 LBD.	Department of Geriatric Medicine (University of Edinburgh). SDCRN (UK).	None.
**Gamified environment (DOREMI)**	Scase, Marandure [45]	Sample 1: *N* = 18: 1° FG: *N* = 9. 5 males. Mean age: 77.0 ± 7.47. Dx: MMSE (29.3 ± 1.00). 2° FG: *N* = 5. 1 male. Mean Age: 74.6 ± 5.46. Dx: MoCA (22.8 ± 1.64). 3° FG: *N* = 4. 1 males. Mean age: 78.5 ± 1.91. Dx: MoCA (22.0 ± 2.45). Sample 2: *N* = 24: Retirement village: *N* = 11. 1 male. Mean age: 75.4 ± 5.14. Dx: MoCA (26.0 ± 2.28). Living separately in a large city: *N* = 13. 1 male. Mean age: 74.9 ± 3.68. Dx: MoCA (24.4 ± 1.19).	Retirement village and city.	None.
**Cognitive Games (DOREMI)**	Scase, Kreiner [46]	Sample 1: *N* = 18: 1° FG: *N* = 9. 5 males. Mean age: 77.0 ± 7.47. Dx: MMSE (29.3 ± 1.00). 2° FG: *N* = 5. 1male. Mean age: 74.6 ± 5.46. Dx: MoCA (22.8 ± 1.64). 3° FG: *N* = 4. 1 males. Mean age: 78.5 ± 1.91. Dx: MoCA (22.0 ± 2.45). Sample 2: N = 25: 3 males. Mean age: 75.0 ± 4.28. Dx: MoCA (24.2 ± 1.71).	Does not specify.	None.
**Puzzle game design**	Gao [22]	*N* = 52: EG: 26. CG: 26. Dx: AD.	Nursing home in a community in Nanjing (China).	None.

Note: AD, Alzheimer’s disease; aMCI, amnestic mild cognitive impairment; CG, control group; DOREMI, Decrease of cOgnitive decline, malnutRition, and sedEntariness by elderly empowerment in lifestyle Management and social Inclusion; Dx; diagnostic; EG, experimental group; FG, focus group; HOP, healthy old people; KLOSCAD, Korean Longitudinal Study on Cognitive Aging and Dementia; LBD, Lewy body dementia; MCI, mild cognitive impairment; MMSE, Mini Mental State Examination; MoCA, Montreal Cognitive Assessment; N; number; naMCI, non-amnestic mild cognitive Impairment; PwD, people with dementia; SDCRN, Scottish Dementia Clinical Research Network; TBI, traumatic brain injury; UK, United Kingdom; USMART, The Ubiquitous Spaced Retrieval-based Memory Advancement and Rehabilitation Training; WoZ, Wizard-of-Oz.

**Table 2 jcm-10-01222-t002:** Methodological design in the development of CCT programs for people with MCI and mild dementia according to ISO9241-210 (21) criteria.

Program/Game	Study	Context of Use					Specify User Requirements				Produce Design Solutions	Design Evaluation				
		Gained Understanding		Specified		Design/Use of the Pre-Prototype	Pre-Prototype Evaluation			Needs and Requirements Identified	Development of the Final Prototype	Methods			Aspects Evaluated	
		OB	Literature Review	Users	CTX		Interviews	Q	OB			OB	Interviews	Q	Usability	UX
MinWii	Benveniste, Jouvelot [37]	X	X	X	X	X	-	-	X	X	NA	NA	NA	NA	NA	NA
MinWii	Boulay, Benveniste [38]	X	X	X	X	NA	NA	NA	NA	NA	X	X	-	X	X	X
COGWEB	Cruz, Pais J. [40]	-	X	X	X	NS	NS	NS	NS	NS	X	-	-	X	X	X
USMART	Han, Oh [41]	-	X	X	X	X	NS	NS	NS	X	X	-	-	X	X	X
LeVer learning platform	Haesner, O’Sullivan [35]	-	X	X	X	X	X	X	-	X	NA	NA	NA	NA	NA	NA
LeVer learning platform	Haesner, Steinert [34]	-	X	X	X	NA	NA	NA	NA	NA	X	-	-	X	X	X
Software “Atama-no-dojo”.	Otsuka, Tanemura [42]	X	X	X	X	NS	NS	NS	NS	NS	X	-	-	X	-	X
X-Top	Ben-Sadoun, Sacco [36]	X	X	X	X	NS	NS	NS	NS	NS	X	-	-	X	X	X
Music ePartner	Peeters, Harbers [43]	-	X	X	X	X	-	X	-	X	X	-	X	-	X	X
Games. WoZ interface	Dethlefs, Milders [44]	-	X	X	X	NS	NS	NS	NS	NS	X	X	X	-	X	X
Gamified environment (DOREMI)	Scase, Marandure [45]	-	X	X	X	X	X	-	-	X	X	-	X	-	-	X
Cognitive Games (DOREMI)	Scase, Kreiner [46]	-	X	X	X	X	X	-	-	X	X	-	X	-	-	X
Puzzle game design.	Gao [22]	-	X	X	X	NS	NS	NS	NS	NS	X	NS	NS	NS	NS	NS

Note: CTX, context; DOREMI, Decrease of cOgnitive decline, malnutRition, and sedEntariness by elderly empowerment in lifestyle Management and social Inclusion; OB, observation; Q, questionnaire; UX, user-experience; NA, did not apply; NS, did not specify; USMART, The Ubiquitous Spaced Retrieval-based Memory Advancement and Rehabilitation Training; WoZ, Wizard-of-Oz.

**Table 3 jcm-10-01222-t003:** User requirements baseline and main outcome measures.

Program	Study	Methodological Design	Instruments for Assessment	Requirements Baseline	Instruments for Assessment of the Final Program	Outcomes Measures
**MinWii**	Benveniste, Jouvelot [37]	UCD.	OB (interactions between user-computer, user-other users, user-caregiver).	Low cognitive and motor requirements. A rewarding UX and hardware. Software and operational simplicity UX.	NA.	NA.
**MinWii**	Boulay, Benveniste [38]	UCD.	NA.	NA.	PSQ.	UX, Usability.
**COGWEB**	Cruz, Pais J. [40]	End-User Interaction Study Design.	NS.	NS.	OQ.	UX. Usability
**USMART**	Han, Oh [41]	UCD.	Evaluation and limitations of a previous program.	Limitations of a previous version: Dispensable presence of the professional and modification in the treatment.	17-item PSQ.	UX. Usability
**LeVer learning platform**	Haesner, O’Sullivan [35]	UCD.	Semi-structured QI. Q	8 factors should be considered when developing an interactive web-based platform for CT. Web design, training, initial evaluation, exercises, feedback, improvement of cognitive performance, communication between users, self-training.	NA.	NA.
**LeVer learning platform**	Haesner, Steinert [34]	UCD.	NA.	NA.	Questions regarding sociodemographic data, computer and tablet use, HB, QoL, technology commitment and, in terms of personality-related characteristics, self-efficacy.	UX. Usability
**Software “Atama-no-dojo”.**	Otsuka, Tanemura [42]	UCD.	NS.	NS.	IQ.	UX.
**X-Top**	Ben-Sadoun, Sacco [36]	UCD.	NS.	NS.	Q.	UX. Usability
**Music ePartner**	Peeters, Harbers [43]	HCD or UCD.	Q to customize the program.	Users preferences. Identification of baseline requirements associated with the role of music in the person’s life, frequency with which they listened to music, if they had played any musical instruments, familiarity with music.	Interview.	UX. Usability
**Games. WoZ interface**	Dethlefs, Milders [44]	UCD.	NS.	NS.	Interview. OB of every session’s interaction between the users and the program (Video recording).	UX. Usability
**Gamified environment (DOREMI)**	Scase, Marandure [45]	UCD.	FG Interviews.	Previous knowledge. Physical characteristics. Customization. Social interaction. Ease of use and understanding. Types of games.	FG Interviews.	UX.
**Cognitive Games (DOREMI)**	Scase, Kreiner [46]	UCD.	FG Interview.	Previous knowledge. Physical characteristics. Customization. Social interaction. Ease of use and understanding. Types of games.	FG Interviews.	UX.
**Puzzle game design**	Gao [22]	HCI.	NS.	NS.	NS.	NS.

Note: CT, cognitive training; DOREMI, Decrease of cOgnitive decline, malnutRition, and sedEntariness by elderly empowerment in lifestyle Management and social Inclusion; FG, focus group; HCD, human-centered design; HCI, human–computer interaction design; IQ, impression questionnaire; NA, did not apply; NS, not specified; OB, observation; OQ, opinion questionnaire; PSQ, patient satisfaction questionnaire; Q, questionnaire; QI, qualitative interview; QoL, quality of life; SMART, Spaced Retrieval-based Memory Advancement and Rehabilitation Training; UCD, user-centered design; UX, user-experience; USMART, The Ubiquitous Spaced Retrieval-based Memory Advancement and Rehabilitation Training; WoZ, Wizard-of-Oz.

## Data Availability

Not applicable.

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
