# Peer review of "Methodological Designs Applied in the Development of Computer-Based Training Programs for the Cognitive Rehabilitation in People with Mild Cognitive Impairment (MCI) and Mild Dementia. Systematic Review"

_jcm, 2021, doi:10.3390/jcm10061222_

Round 1

Reviewer 1 Report

The paper addresses a relevant and innovative research question. The report is well written, the method section is detailed and in compliance with PRISMA standards.

Two small issues should be addressed:

1) As the authors state themselves in the 'limitations' section, their search strategy seems somewhat limited (only two databases, none of which explicitly address technological works; only one day of searching?). The authors should give the reasons for this strategy (apart form economic reasons?) in the methods or limitations section.

2) the recommendations/conclusions sections are too vague and should be presented in a more structured way (e.g., including a list of aspects that developers of programs should pay attention to) so that the reader can see the implications and relevance of this review more clearly.

Author Response

Thank you very much for considering the review of manuscript "Methodological Designs Applied in the Development of Computer-based Training Programs for the Cognitive Rehabilitation in People with Mild Cognitive Impairment (MCI) and Mild Dementia. Systematic Review" (jcm-1128618). Below, I offer an answer to each of the comments made.

Reviewer 1

Two small issues should be addressed:

1) As the authors state themselves in the 'limitations' section, their search strategy seems somewhat limited (only two databases, none of which explicitly address technological works; only one day of searching?). The authors should give the reasons for this strategy (apart form economic reasons?) in the methods or limitations section.

Author: Corrected (limitations session, line 848, page 19).

2) the recommendations/conclusions sections are too vague and should be presented in a more structured way (e.g., including a list of aspects that developers of programs should pay attention to) so that the reader can see the implications and relevance of this review more clearly.

Author: Corrected (conclusions session, line 920-953, page 20-21).

Thank you for your consideration of this manuscript.

Sincerely,

Angie A. Diaz-Baquero

Reviewer 2 Report

Authors reviewed the methodological designs of the published computer-based training programs for the cognitively impaired people based on ISO9241-210.

In abstract, authors should include the period of search, 2000-2019, for the relevant articles for this review.  

conclusion is generally placed in the abstract rather than discussion.  

In Table 1, it may be easier to understand to have a note (or separate column) indicating whether each study employed 2 samples or not.

Table 1 needs to be summarized better. There are unnecessarily too many abbreviations and unexplained superscripts. Author may include more important stats in Table 1 and send others to a supplementary table if possible. 

Table 2, please explain the meaning of blank. Is it also "did not apply" or not known?

Two studies did not develop program (Benveniste (37), Haesner, O'Sullivan (35)). While eligibility for inclusion didn't mention about this, I would like authors to discuss why these two studies were considered.  

Finally, I would like authors to discuss significance of employing two samples (one group is employed from inception, and another independent group for design evaluation after development).

Author Response

Thank you very much for considering the review of manuscript "Methodological Designs Applied in the Development of Computer-based Training Programs for the Cognitive Rehabilitation in People with Mild Cognitive Impairment (MCI) and Mild Dementia. Systematic Review" (jcm-1128618). Below, I offer an answer to each of the comments made.

In abstract, authors should include the period of search, 2000-2019, for the relevant articles for this review.  

Author: Corrected (Abstract session, line 26, page 1).

Conclusion is generally placed in the abstract rather than discussion.  

Author: Corrected (Abstract session, line 33-37, page 1).

In Table 1, it may be easier to understand to have a note (or separate column) indicating whether each study employed 2 samples or not.

Author: Corrected (Materials and Methods session, line 206, page 6-8).

Table 1 needs to be summarized better. There are unnecessarily too many abbreviations and unexplained superscripts. Author may include more important stats in Table 1 and send others to a supplementary table if possible. 

Author: Corrected (Materials and Methods session, line 206, page 6-8).

Table 2, please explain the meaning of blank. Is it also "did not apply" or not known?

Author: Corrected (Results session, line 662, page 11-12).

Two studies did not develop program (Benveniste (37), Haesner, O'Sullivan (35)). While eligibility for inclusion didn't mention about this, I would like authors to discuss why these two studies were considered.  

Author: (Materials and Methods session, line 187-194, page 5) and (Results session, line 550-558, page 8)

Finally, I would like authors to discuss significance of employing two samples (one group is employed from inception, and another independent group for design evaluation after development).

Author: Corrected (Discussion session, line 800-803, page 15).

Thank you for your consideration of this manuscript.

Sincerely,

Angie A. Diaz-Baquero